# Growth hormone increases regulator of calcineurin 1-4 (*Rcan1-4*) mRNA through c-JUN in rat liver

Tomoko Nakata, Yuka Toyoshima, Takashi Yagi, Harumi Katsumata, Reiko Tokita, Shiro Minami*

Department of Bioregulation, Institute for Advanced Medical Sciences, Nippon Medical School, Kawasaki, Japan

* shirom@nms.ac.jp

**Data Availability Statement:** All relevant data are within the manuscript and its Supporting Information files.

## Abstract

Growth hormone (GH) activates multiple signal transduction pathways. To investigate these pathways, we identified novel genes whose transcription was induced by GH in the liver of hypophysectomized (HPX) rats using the suppression subtractive hybridization technique. We found that regulator of calcineurin 1 (*Rcan1*) mRNA was upregulated by GH administration. RCAN1 regulates the activity of calcineurin, a Ca/calmodulin-dependent phosphatase. *Rcan1* encodes two major transcripts, *Rcan1-1* and *Rcan1-4*, resulting from differential promoter use and first exon choice. We found that a single injection of GH increased the levels of *Rcan1-4* mRNA and RCAN1-4 protein transiently, but did not increase *Rcan1-1* mRNA in HPX rat liver. Then the molecular mechanism of GH to induce *Rcan1-4* transcription was examined in rat hepatoma H4IIE cells. Experiments using inhibitors suggested that c-JUN N-terminal kinase was required for the induction of *Rcan1-4* mRNA by GH. GH increased the levels of phosphorylated c-JUN protein and *c-Jun* mRNA in HPX rat liver. The luciferase and electrophoretic mobility shift assays showed that c-JUN upregulated *Rcan1-4* mRNA by binding to the cAMP-responsive element in the upstream of *Rcan1* exon 4. These results indicate that GH activates c-JUN to affect the activity of calcineurin by the induction of *Rcan1-4* in rat liver.

## Introduction

Growth hormone (GH) is not only a critical hormone for growth but also plays a role in numerous important physiological processes, such as lipolysis, gluconeogenesis, and protein synthesis [1, 2]. In particular, the effects of GH on transcription in the liver, where GH receptors are abundantly expressed, have been extensively studied [3, 4]. The actions of GH are mediated directly by its own receptor and indirectly by production of insulin-like growth factor 1 (IGF-1) [5]. Because many unknown genes could be regulated by GH, we tried to identify new genes whose transcription was induced by GH in the liver of hypophysectomized (HPX) rats using the suppression subtractive hybridization (SSH) technique [6]. We found that GH increased regulator of calcineurin 1 (*Rcan1*) mRNA in HPX rat liver.

**Funding:** The author(s) received no specific funding for this work.

**Competing interests:** The authors have declared that no competing interests exist.

RCAN1, initially referred to DSCR1, MCIP1, or Adapt78, was first identified as a Down syndrome critical region gene on human chromosome 21q22 [7]. RCAN1 interacts with the catalytic subunit of calcineurin and acts as either an inhibitor or a facilitator of calcineurin [8–11]. Calcineurin is a Ca/calmodulin-dependent phosphatase that plays a role in numerous processes including T cell activation [12, 13], skeletal [14] and cardiac myocytes growth [15], induction of angiogenesis [7, 16], and gluconeogenesis [17]. Calcineurin dephosphorylates the nuclear factor of activated T cells (NFAT), enabling it to translocate to the nucleus and act as a transcription factor [13, 18]. *Rcan1* encodes two major transcripts, *Rcan1-1* and *Rcan1-4*, which are expressed by different promoter. *Rcan1-4* is upregulated by calcineurin. The promoter region of *Rcan1-4* includes a cluster of consensus binding sites for NFAT [19].

GH signals mainly through activation of the Janus kinase (JAK)-signal transducer and activator of transcription (STAT) pathway. STAT5B regulates the expression of target genes associated with several physiological processes, including body growth, the cell cycle, and lipid, bile acid, steroid, and drug metabolism. Disruption of STAT5 signaling causes fatty liver, fibrosis, and hepatocellular carcinoma (HCC) [20]. Mitogen-activated protein kinase (MAPK)/extracellular signal-regulated protein kinase (ERK) kinase (MEK) pathways are also important. But only a few transcripts that ERK phosphorylated by GH regulates are known [21]. It was suggested that GH also activates c-JUN N-terminal kinase (JNK) [22]. JNK is known to phosphorylate c-JUN at Ser63 and Ser73 and their phosphorylation activates c-JUN [23]. c-JUN binds to the sequences designated as ATF/CRE, TGACGTCA and AP-1/TRE, TGAG/CTCA. ATF/CRE is defined as the activating transcription factor (ATF) binding site or the cAMP responsive element (CRE). AP-1/TRE is the activator protein 1 (AP-1) binding site or the 12-O-tetradecanoylphorbol-13-acetate (TPA) response element (TRE). Whereas c-JUN homodimer and c-JUN/c-FOS heterodimer bind to TRE, c-JUN/c-FOS heterodimer binds more efficiently to TRE than c-JUN homodimer does [24]. c-JUN homodimer can bind to CRE. c-JUN also forms a heterodimer with ATF, and c-JUN/ATF heterodimer is considered to bind to CRE with a higher affinity than to the TRE site [25, 26]. Active c-JUN autoregulates its promoter by binding to CRE-like sequence TGACATCA [27].

The purpose of this study was to investigate the molecular mechanism of *Rcan1-4* expression induced by GH.

## Materials and methods

### Materials

Recombinant rat GH (rGH) and prolactin (PRL) were supplied by the National Hormone and Pituitary Program, National Institute of Diabetes and Digestive and Kidney Diseases (NHPP-NIDDK). Recombinant human GH (hGH) was a generous gift from Novo Nordisk Pharma (Tokyo, Japan). Radionucleotide [$\alpha$-$^{32}$P] CTP (3000 Ci/mmol) was from PerkinElmer (Waltham, MA).

### Methods

**Animal preparation and *in vivo* experiments.** Animal preparation and *in vivo* experiments were described previously [28]. All experimental protocols were reviewed and approved in advance by the Experimental Animal Ethics Committee of Nippon Medical School (Approval Number 18–140). Hypophysectomy (HPX) was performed *via* the parapharyngeal approach, and HPX rats received a daily subcutaneous injection of dexamethasone phosphate (20 μg/kg) and L-thyroxine (20 μg/kg). For bolus injection of GH, three days prior to GH administration, rats were fitted with an indwelling right atrial cannula *via* the external jugular vein under ketamine-xylazine anesthesia for systemic treatment with either recombinant rGH

or vehicle. Vehicle contained 30 mM NaHCO₃, 0.15 M NaCl, and 100 μg/ml rat albumin. On the day of the experiment, rats were sacrificed by decapitation at various times following intravenous injection of rGH or vehicle. Tissue samples were removed from the liver, immediately frozen in liquid nitrogen, and then stored at -80˚C until analysis.

**PCR select cDNA subtraction, cloning, and sequencing.** Genes induced by GH in HPX rat livers were determined by the SSH technique [6] using the PCR-Select cDNA Subtraction Kit (Clontech Laboratories, Inc., Palo Alto, CA) as described previously [28]. One microgram of total RNA from GH or vehicle-treated rat liver was reverse transcribed using the SMART PCR cDNA Synthesis Kit (Clontech Laboratories, Inc.). Two subtractive libraries were prepared as follows. A forward subtractive library was prepared by subtracting vehicle-treated adaptor-free cDNAs from GH-treated adaptor-ligated cDNAs, and a reverse subtractive library was generated by subtracting GH-treated adaptor-free cDNAs from vehicle-treated adaptor-ligated cDNAs. The subtracted cDNAs were cloned into a pBluescriptSK+ vector (Toyobo, Osaka, Japan), and an aliquot of this library was plated on nylon membranes. Forward or reverse subtracted cDNA probes were labeled with digoxigenin (DIG), and then the membranes were screened by differential hybridization by following the instructions in the DIG High Prime DNA Labeling Kit (Roche Applied Science, Mannheim, Germany). Clones that hybridized specifically to the forward probe were screened by virtual northern blot analysis. Virtual northern blots were blots of full-length cDNA, made by PCR amplification of the primary cDNA [29]. cDNAs were amplified with Advantage 2 Polymerase Mix. Cycle conditions were 95˚C for 1 min, then 17 cycles of 95˚C for 5 sec, 65˚C for 5 sec, and 68˚C for 6 min in a Gene Amp PCR System 9700 (Applied Biosystems, Foster City, CA). The PCR product was separated on a 1.2% agarose gel, denatured and subsequently blotted onto a nylon membrane in a conventional southern transfer and UV-crosslinked. Each clone was labeled with PCR DIG Probe Synthesis Kit (Roche). Prehybridization, hybridization, washing and detection were performed according to the manufacturers' protocols. Clones that showed differential expression were sequenced by the BigDye Terminator Cycle Sequencing method using a 377 ABI Prism automated DNA sequencer (PerkinElmer) according to the manufacturers' protocols. The sequences obtained were compared to those in GenBank using BLAT in the UCSC Genome Browser.

**RNA isolation and northern blotting.** Total RNA and Poly(A) RNA were isolated as described [28]. 7.5 μg total RNA or 0.5 μg poly(A) RNA was electrophoresed on a 1% agarose and 6% formamide gel, transferred to a Hybond-N+ membrane (GE Healthcare UK Ltd., Buckinghamshire, UK), and hybridized to radiolabeled complementary RNA (cRNA) probes derived from cDNAs as described previously [30].

**Real time quantitative RT-PCR (qPCR).** cDNA were prepared as described [28]. The oligonucleotides utilized are listed in Table 1. qPCR was performed with SYBR-Premix Ex Taq (Takara, Shiga, Japan) to a final volume of 10 μl. qPCR measurements were performed on an ABI Prism 7700 sequence detector system (Applied Biosystems). Final measurements were normalized by the target mRNA/*Gapdh* average value for all samples.

**Treatment of cells.** H4IIE cells expressing the *Ghr* mRNA (H4IIE GHR) were constructed previously [28], and stored in alpha modified Eagle's medium containing 10% fetal calf serum and 10% DMSO in liquid nitrogen freezer. H4IIE GHR cells at 16–20 passages were used in experiments. Prior to treatment, cells were deprived of serum for 16–30 h in alpha modified Eagle's medium. Cells were preincubated with a certain inhibitor or vehicle (DMSO, 0.1% final concentration) for 1 h prior to treatment with GH. The final concentrations of the MAPK/ERK kinase (MEK) inhibitor PD98059 (Cell Signaling Technology, Danvers, MA), JNK inhibitor SP600125 (Calbiochem, San Diego, CA), and cyclosporine (CHX) (Calbiochem) were 50 μM, 10 μM, and 1 μM, respectively, and hGH was used at a concentration of 500 ng/ml.

**Table 1. Sequences of oligonucleotides.**

| qPCR | |
| --- | --- |
| Rcan1-1 forward | 5'-ACTGCGAGATGGAGGAGGTG-3' |
| Rcan1-1 reverse | 5'-CGTCCTGAAGAGGGATTCAAA-3' |
| Rcan1-4 forward | 5'-CTTGGGCTTGACTGCGAGTGAGTG-3' |
| Rcan1-4 reverse | 5'-CCCTGGTCTCACTTTCGCTG-3' |
| c-Jun forward | 5'-CGGGCTGTTCATCTGTTTGT-3' |
| c-Jun reverse | 5'-CCGGGACTTGTGAGCTTCTT-3' |
| Gapdh forward | 5'-GATGCTGGTGCTGAGTATGTCG-3' |
| Gapdh reverse | 5'-TGGTGCAGGATGCATTGCTG-3' |
| EMSA | |
| CRE | 5'-ATGACTAGGGTGTTGACGTCACCTCTTTCCAGTA-3' |
| TRE | 5'-GGCACGCGGGACTTGACTCAGGAATTTGCTGT-3' |
| STAT5B | 5'-GGACTTCTTGGAATTAAGGGA-3' |

**Cell culture and transfections (luciferase).** Reporter plasmids were constructed by inserting a 5' upstream fragment of *Rcan1-4* into pGL4.12-basic (Promega Corporation, Madison, WI). The rat *c-Jun* expression plasmid was constructed by inserting cDNA of rat *c-Jun* into pcDNA3.1. For transient transfections, cells were seeded in 24-well plates at a density of $8 \times 10^4$ cells/well. Fugene HD transfection reagent was used at a ratio of 3:1 of Fugene HD/DNA (vol/wt), as described in the manufacturer's protocol. Each well received 250 ng luciferase reporter plasmid and 100 ng *c-Jun* expression plasmid. pRL-tk-Luc plasmid (*Renilla* luciferase; 10 ng DNA) was included as an internal control for transfection efficiency. pcDNA3.1 DNA was used to adjust the total to 500 ng of DNA per well. Forty-eight hours after transfection, total cell extracts were prepared using 1x lysis buffer (Promega) for measuring luciferase activities. Firefly and *Renilla* luciferase activities were measured using a Dual Reporter Assay System (Promega) and Sirius luminometer (Berthold Detection Systems, Pforzheim, Germany). Firefly luciferase activity values were divided by *Renilla* luciferase activity values to obtain normalized luciferase activities (mean ± standard error (SE) values for $n = 3$ separate transfections).

**Protein extraction and western blotting.** Isolation of cytoplasmic and nuclear proteins has been described previously [28]. Protein extracts were electrophoresed through SDS polyacrylamide gels. Gels were electrophoretically transferred to Immobilon PVDF membranes (Millipore, Bedford, MA) and reacted with the indicated antibodies. Immunoreactive proteins were visualized by the chemiluminescence detection system ImmunoStar LD (Wako, Osaka, Japan). The following antibodies were used: anti-RCAN1 and anti-beta-actin antibodies (Sigma-Aldrich, St. Louis, MO), anti-phospho-c-JUN (Ser63), and anti-c-JUN antibodies (Cell Signaling Technology), and anti-TATA binding protein (TBP) antibody (Abcam, Cambridge, UK).

**Electrophoretic mobility shift assay (EMSA).** EMSA was performed as described [28]. Nuclear extracts of the control and *c-Jun* expression plasmid transfected HeLa cells were added to binding buffer, and the mixture was incubated for 10 min at 25°C. Digoxigenin-labeled double-stranded oligonucleotide (Table 1) was added in the absence or presence of an unlabeled competitor, and the binding mixture was incubated for 30 min at 25°C. For some experiments, anti-c-JUN or anti-STAT5B antibodies (Santa Cruz Biotechnology, Inc. Santa Cruz, CA) were added and incubated for 10 min. The mixture was kept on ice for 10 min and loaded onto the 4% polyacrylamide gels in 0.5x TBE. The gel was run in 0.5x TBE buffer at 100 V for 70 min. After electrophoresis, the gel was transferred onto a Hybond-N+ membrane (GE Healthcare UK Ltd.) and cross-linked by UV. Detection was performed as described [28].

**Statistical analysis.** All results are expressed as the mean ± SE for each group of rats. Main effects and interactions were analyzed by two-way ANOVA followed by Bonferroni's test. Comparison between groups of RCAN1-4 proteins was performed by Student's *t* test. Comparison between groups of *Rcan1-4* mRNAs of PRL and vehicle treated rat liver was performed by Welch's *t* test. The level of significance was indicated as * for GH-treated vs. vehicle-treated rats or cells, # for GH-treated vs. untreated rats, and $ for c-jun-expressed vs. pcDNA-expressed cells. A single symbol (*, #, or $) indicates significance at $p < 0.05$, and a double symbol indicates $p < 0.01$.

## Results

### GH induces *Rcan1-4* mRNA expression in rat liver

To identify genes induced by GH in HPX rat liver, we used the SSH technique as described in Materials and Methods. We prepared RNA from HPX rat liver at 1 h after intravenous injection of GH or vehicle. From the forward subtracted library (subtracting vehicle-treated cDNAs from GH-treated cDNAs), 702 colonies were isolated. 48 clones were hybridized with only forward subtracted probe but not reverse subtractive probe (subtracting GH-treated cDNAs from vehicle-treated cDNAs). To identify the clones that were really upregulated by GH, they were analyzed by virtual northern blot and 8 clones that were upregulated by GH were sequenced (S1 Table). mRNA levels of these genes were further determined by northern blotting to confirm that GH regulated the expression of these genes. We found that the levels of *Rcan1*, *Xbp1*, *Nampt*, and *Leap2* mRNAs were much higher at 1 h after GH injection than those after vehicle injection (Fig 1A). These genes have not been described as GH responsive genes [31], although Wauthier and Waxman indicated in microarray data that *Rcan1* and *Nampt* mRNAs were increased by GH [32]. We have studied *Rcan1* and *Xbp1* in detail because GH increased the levels of RCAN1 and XBP1 proteins. We reported about *Xbp1* previously [28]. When HPX rats were treated with vehicle, *Rcan1* mRNAs were hardly detectable (Fig 1B). When HPX rats were treated with increasing doses of GH (8–1000 μg/kg), 3.0-kb and 2.0-kb *Rcan1* mRNAs were detected at 200 μg/kg and increased more at 1000 μg/kg (Fig 1B). Because *Rcan1* mRNAs were increased at 200 μg/kg of GH, we examined the time course of *Rcan1* mRNA expression at this concentration. In the absence of GH, northern blot analysis detected both *Rcan1* mRNAs scarcely. Both *Rcan1* mRNAs increased at 0.5–2 h after GH injection and then decreased at 4 h (Fig 1C).

*Rcan1* encodes two major transcripts, *Rcan1-1* and *Rcan1-4*. *Rcan1-1* transcript, which contains exons 1, 5, 6, and 7, is regulated by a 5' promoter, whereas *Rcan1-4* transcript, which contains exons 4, 5, 6, and 7, is regulated by a promoter located upstream of exon 4 (Fig 2A). We performed qPCR using specific primers for exons 1 and 4 to examine which mRNA is increased by injection of GH in the HPX male rat liver. Upon intravenous injection of GH, the level of *Rcan1-1* mRNA was not increased during observation period (Fig 2B). In contrast, GH injection caused a vigorous increase of the level of *Rcan1-4* mRNA to 5-fold at 0.5 h and to about 14-fold at 2 h, which diminished to basal level at 4 h (Fig 2C). These results indicate that *Rcan1-4* mRNA is induced by GH. Because qPCR revealed that *Rcan1-1* mRNA was not significantly increased by GH, both 3.0-kb and 2.0-kb mRNAs which were increased by GH as shown in Fig 1C are likely to be *Rcan1-4* mRNA. Injection of the same amount of PRL did not significantly increase *Rcan1-4* mRNA, which indicated that GH but not PRL increased *Rcan1-4* mRNA (Fig 2D).

### GH increases RCAN1-4 protein in rat liver

We investigated whether RCAN1-4 protein increased when GH was injected intravenously. Immunoblotting using anti-RCAN1 antibody raised against the C-terminus of human

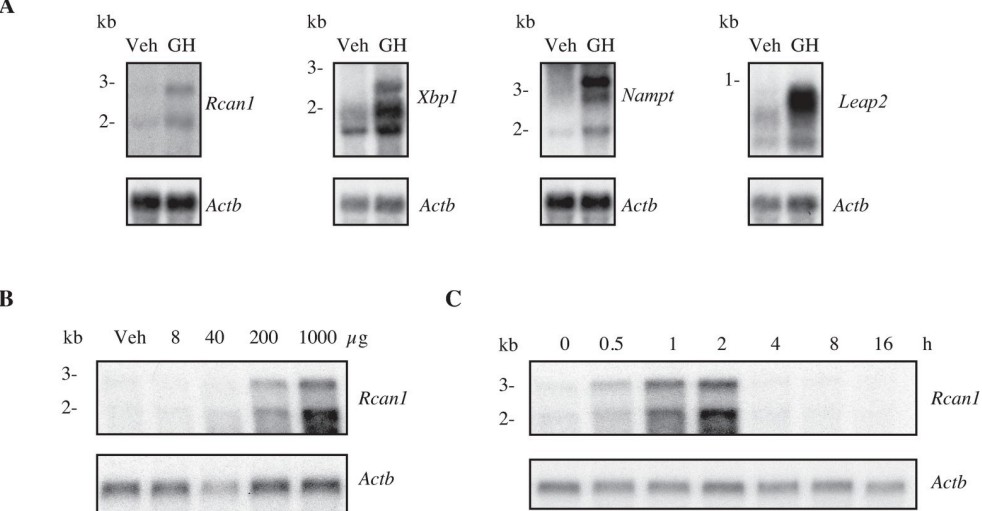

**Fig 1. GH increases *Rcan1* mRNA in HPX rat liver.** Northern blot analysis of mRNAs in HPX rat liver. A. Total RNA (7.5 μg/lane) isolated from HPX rat livers at 1 h after administration of vehicle or hGH (1000 μg/kg) were hybridized with cRNA probes for *Rcan1*, *Xbp1*, *Nampt*, *Leap2* and *Actb* (β-actin). B and C. Poly(A) RNA (0.5 μg/lane) was hybridized with cRNA probes for *Rcan1* and *Actb* (β-actin). Poly(A) RNA isolated from HPX rat livers at 2 h after administration of increasing doses of rGH (8–1000 μg/kg) (B). Poly(A) RNA isolated from HPX rat livers prior to or following rGH treatment (200 μg/kg) for the indicated time points (C). Results are representative of 4 independent experiments.

RCAN1 revealed that two protein bands corresponding to about 28 kDa increased at 1 h, reached a maximum at 2 h, and declined to basal level at 4 h (Fig 2E). These profiles are consistent with those of *Rcan1-4* mRNA as described in Fig 2C. Because both hyper-phosphorylated and hypo-phosphorylated RCAN1-4 proteins at about 29 kDa were reported in U251 astroglioma and HeLa cells [33, 34], the two bands may reflect the difference of phosphorylation states of RCAN1-4 protein. We also investigated RCAN1-4 protein levels in normal and HPX rat livers of both sexes. RCAN1-4 protein levels were more abundant in normal rat livers of both sexes than HPX rat livers (Fig 2F). Although many GH regulated proteins are sexually dimorphic in rat liver, there was no significant difference in RCAN1-4 protein levels between males and females (Fig 2G).

## Signaling pathways that regulate *Rcan1-4* expression upon GH treatment

To investigate the molecular mechanism by which GH promotes an increase of *Rcan1-4* mRNA, we constructed H4IIE cells that stably expressed the GH receptor, GHR (H4IIE GHR). When H4IIE GHR cells were treated with GH, *Rcan1-4* mRNA levels increased after 1–2 h (Fig 3A). Because *Rcan1-4* mRNA is regulated by the calcineurin-NFAT signaling pathway [35], we examined the effects of cyclosporine A (CsA), an inhibitor of the phosphatase activity of calcineurin, to an increase of *Rcan1-4* mRNA levels upon GH treatment. CsA did not affect the levels of GH-induced *Rcan1-4* mRNA (Fig 3B), suggesting that GH induces the expression of *Rcan1-4* mRNA through a calcineurin-NFAT independent pathway. Then, we examined whether the MEK-ERK pathway or JNK pathway contributes to GH-stimulated *Rcan1-4* expression. The MEK inhibitor PD98059 did not affect the GH-induced *Rcan1-4* mRNA levels (Fig 3C). In contrast, the JNK inhibitor SP600125 reduced the levels of GH-induced *Rcan1-4* mRNA (Fig 3C). These results suggest that JNK but not MEK-ERK plays a role in the increase of *Rcan1-4* mRNA levels by GH.

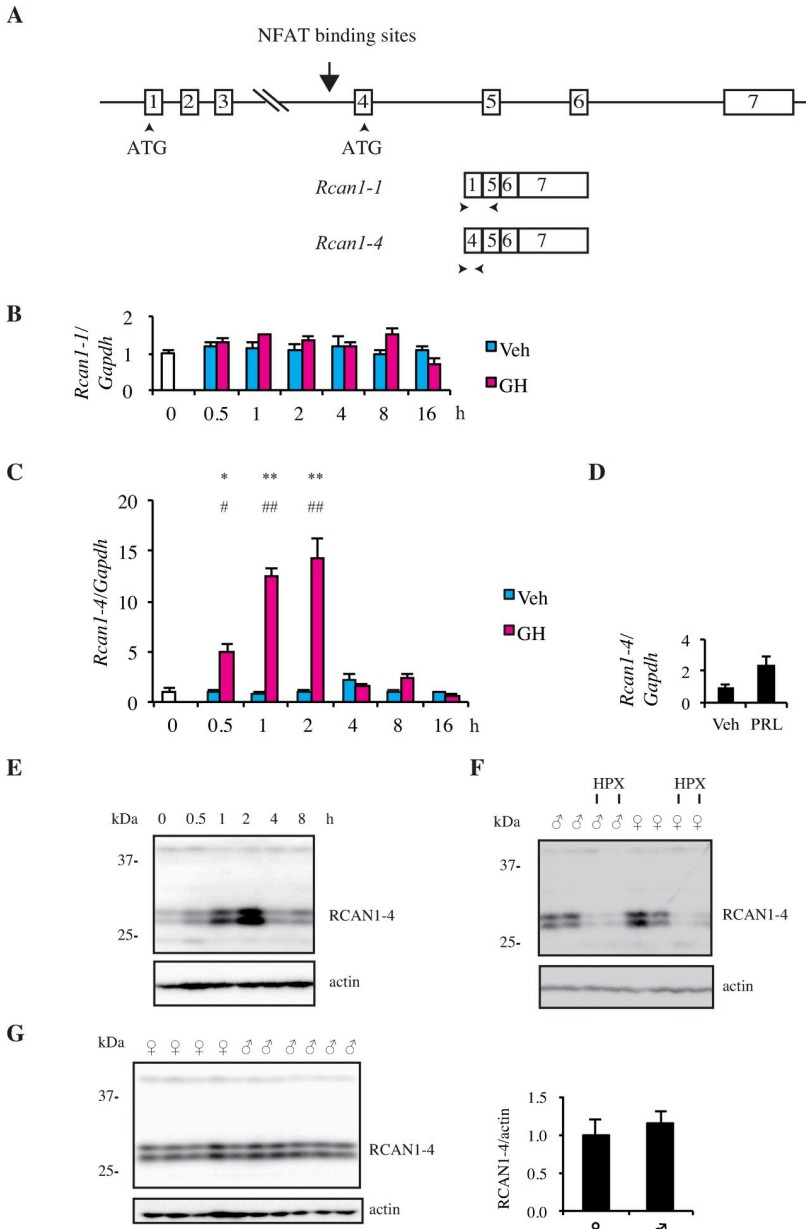

**Fig 2. GH increases *Rcan1-4* mRNA and RCAN1-4 protein in HPX rat liver.** A. Structures of the *Rcan1* gene and *Rcan1-1* and *Rcan1-4* mRNAs. A vertical arrow indicates putative NFAT binding sites. Arrowheads indicate the primers used to detect *Rcan1-1* and *Rcan1-4* mRNAs. B and C. mRNA levels of *Rcan1-1* (B) and *Rcan1-4* (C) isolated from HPX rat livers prior to or following GH (200 μg/kg) or vehicle treatment for the indicated time points were determined by qPCR analysis. Values were normalized to *Gapdh* as the reference gene and were presented relative to the averaged level of untreated HPX male liver, which was set to 1. Data are the mean ± SE of 4 rats. **$p < 0.01$, and *$p < 0.05$ for GH-treated vs. vehicle-treated rats. ## $p < 0.01$, and # $p < 0.05$ for GH-treated vs. untreated rats. D. mRNA levels of *Rcan1-4* isolated from HPX rat livers following PRL (200 μg/kg) or vehicle treatment after 2 hr were determined by qPCR analysis. Values were normalized to *Gapdh* as the reference gene and were presented relative to the averaged level of vehicle treated HPX male liver, which was set to 1. Data are the mean ± SE of 5 rats. E. Cytoplasmic extracts from HPX rat livers prior to or following administration of GH (200 μg/kg) for the indicated time points were immunoblotted using anti-RCAN1 and anti-actin antibodies. F. Cytoplasmic extracts from rat livers of normal males, HPX males, normal females, and HPX females were immunoblotted using anti-RCAN1 and anti-actin antibodies. G. Cytoplasmic extracts from rat livers of normal males and normal females were immunoblotted using anti-RCAN1 and anti-actin antibodies. Right panel: Protein levels of male and female RCAN1/actin are quantified.

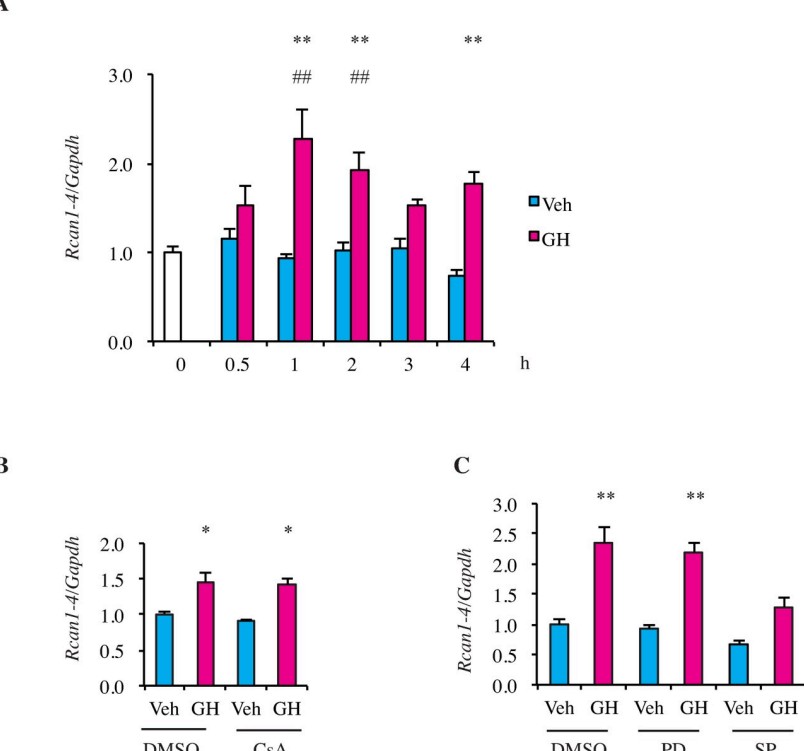

**Fig 3. Analysis of signaling pathways regulating *Rcan1-4* expression.** A. H4IIE GHR cells were treated with hGH (500 ng/ml) or vehicle at the indicated time points. *Rcan1-4* mRNA levels were determined by qPCR analysis. RNA levels were normalized to *Gapdh* as the reference gene and were presented relative to the averaged non-treated level, which was set to 1. B and C. H4IIE GHR cells were preincubated for 1 h with DMSO, CsA (B) or PD98059, SP600125 (C), followed by hGH or vehicle for 1 h. *Rcan1-4* mRNA levels were determined by qPCR analysis and normalized to the *Gapdh* content of each samples and were presented relative to the averaged level of DMSO and vehicle-treated cell, which was set to 1. Data shown are the mean ± SE of 4 experiments. **$p < 0.01$, and *$p < 0.05$ for GH vs. vehicle. ## $p < 0.01$ for GH-treated vs. untreated cells.

## GH increases phosphorylated c-JUN protein and *c-Jun* mRNA

JNK is known to phosphorylate c-JUN at Ser63 and Ser73 to activate c-JUN [23]. We investigated whether injection of GH in HPX rats increased the level of phosphorylated c-JUN. Immunoblotting with anti-c-JUN antibody showed that c-JUN protein increased after 1–2 h and declined to basal level at 4 h (Fig 4A). Ser63-phosphorylated c-JUN protein increased and decreased slightly earlier than those of total c-JUN protein. These results suggest that GH activates JNK and increases the level of phosphorylated c-JUN protein.

Next, we investigated the levels of *c-Jun* mRNA in HPX rat liver. A single injection of GH increased *c-Jun* mRNA at 0.5 h, which reached a maximum at 1 h of about 20-fold of the vehicle-treated liver. The *c-Jun* mRNA level sharply declined to basal level at 4 h (Fig 4B).

## GH induces *Rcan1-4* expression through c-JUN

Next, we investigated the mechanism by which c-JUN increases *Rcan1-4* mRNA. We searched for putative transcription factor binding sites upstream of the *Rcan1-4* transcription initiation site using the TRANSFAC database [36]. In addition to NFAT binding sites, we found one putative binding site designated as ATF/CRE and one as AP-1/TRE (Fig 5A). The DNA fragment from -458 bp 5'-upstream of the *Rcan1-4* transcriptional initiation site was used to drive

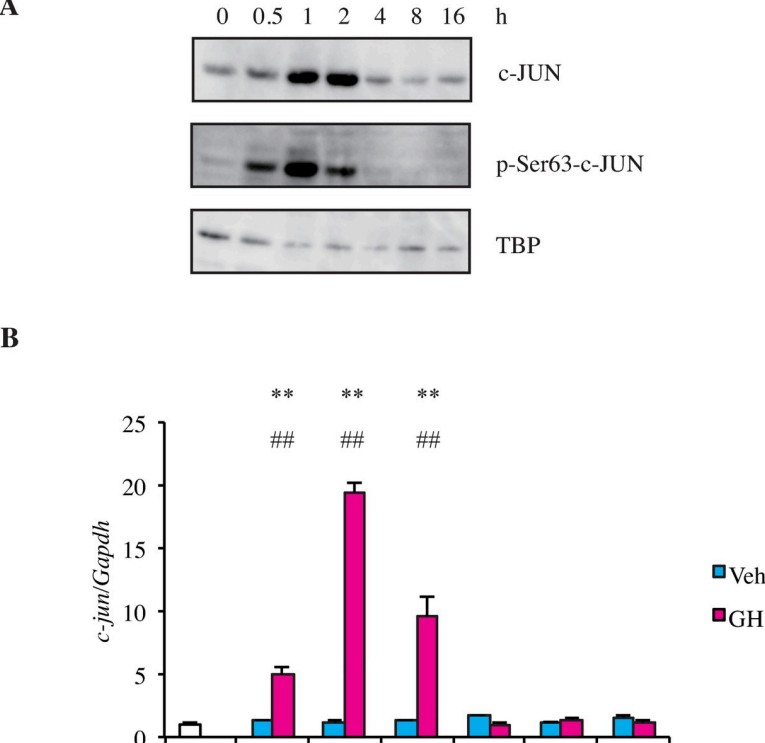

**Fig 4. GH increases active c-JUN protein and *c-Jun* mRNA in HPX rat liver.** A. Nuclear extracts isolated from HPX rat livers prior to or following GH treatment (200 μg/kg) at the indicated time points were immunoblotted using anti-c-JUN, anti-p-Ser63-c-JUN, and anti-TBP antibodies. B. *c-Jun* mRNA levels of HPX rat liver at the indicated time points after *iv* administration of GH or vehicle were determined by qPCR analysis. Values were normalized to *Gapdh* as the reference gene and were presented relative to the averaged untreated HPX male liver level, which was set to 1. **$p < 0.01$ for GH vs. vehicle. ## $p < 0.01$ for GH-treated vs. untreated rats.

expression of the firefly luciferase reporter gene using H4IIE GHR cells (Fig 5A). When the pGL4.12-basic plasmid, which has some consensus transcription factor binding sites including AP-1/TRE, was co-transfected with a *c-Jun* expression plasmid, relative luciferase activity increased but remained at a low level (Fig 5B). In contrast, co-transfection of the construct carrying the 5'-upstream fragment of *Rcan1-4* with the *c-Jun* expression plasmid caused a vigorous increase in the luciferase activity compared to the expression vector pcDNA3.1 (Fig 5B). These results indicate that c-JUN activates the *Rcan1-4* promoter.

To examine whether c-JUN binds to either or both of the putative CRE and TRE binding sites, we performed EMSA using nuclear extracts prepared from HeLa cells transfected with *c-Jun* expression plasmid. *c-Jun* expression caused a shift of the CRE DNA probe, generating a slowly migrating band (shown as "shift" in Fig 5C, lane 2 compared to lane 1). This band was likely to be a complex containing c-JUN protein, because the band was supershifted by anti-c-JUN antibody (Fig 5C, lanes 4) but not by a negative control, anti-STAT5B antibody (Fig 5C, lanes 6). The complex was diminished by an excess amount of unlabeled CRE binding probe (Fig 5C, lanes 7) but not by the negative control, STAT5B binding probe (Fig 5C, lanes 8), showing that c-JUN binds to CRE. In contrast, *c-Jun* expression did not cause significant changes in migration of the TRE DNA probe (Fig 5C, lanes 9–10). These results suggest that c-JUN binds to CRE at the *Rcan1-4* promoter and plays a role in induction of *Rcan1-4* expression.

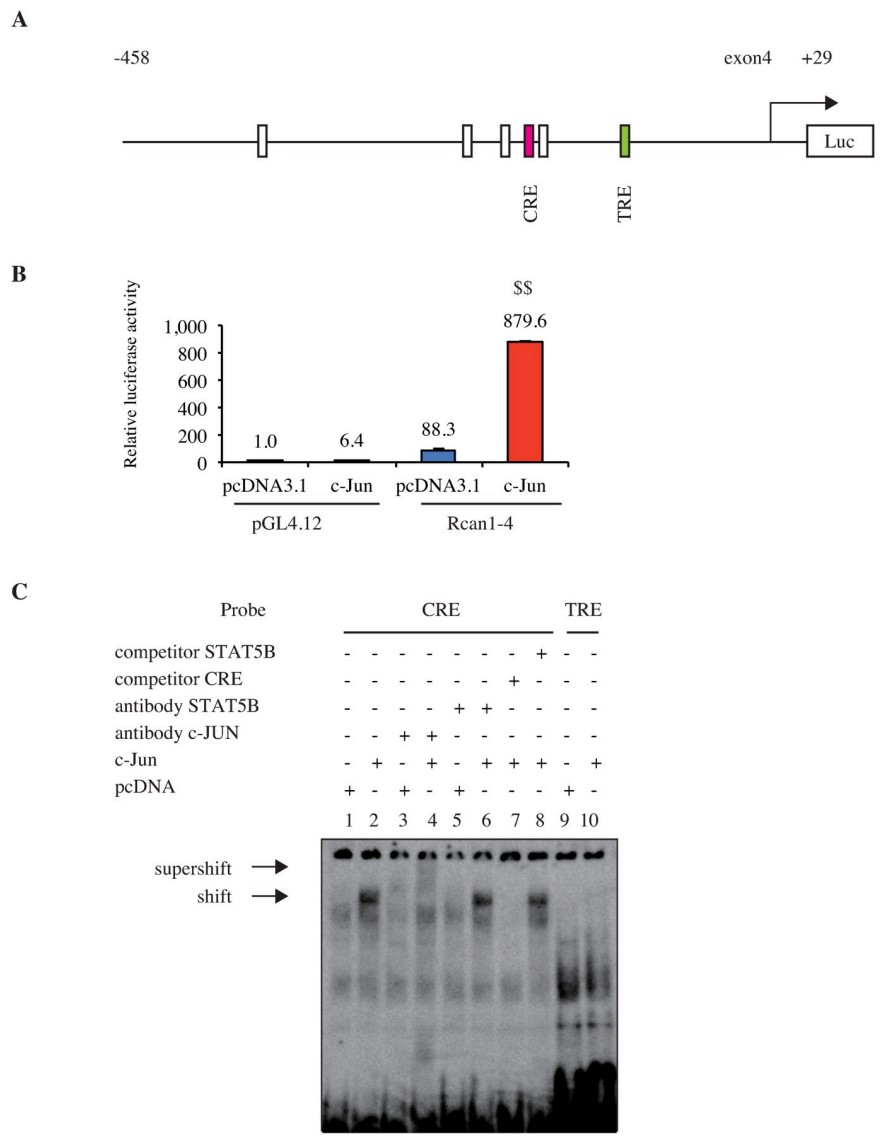

**Fig 5. c-JUN activates *Rcan1-4* promoter by binding CRE.** A. Scheme of the rat *Rcan1-4* promoter fused to the luciferase gene. Rectangles indicate putative c-JUN binding elements, CRE (TGACGTCA, red) and TRE (TGACTCA, green), and NFAT binding elements (GGAAA, white). B. H4IIE GHR cells were transfected with pGL4.12 or *Rcan1-4-Luc* along with pcDNA3.1 or *c-Jun* expression plasmid. The averaged luciferase activity of cells transfected with pGL4.12 and pcDNA3.1 was set to 1. Relative luciferase activity is shown above the bar. Data shown are the mean ± SE of 3 experiments. $ $ *p* < 0.01 for c-jun- expressed vs. pcDNA-expressed cells. C. Nuclear extracts from HeLa cells transfected with a control plasmid (lanes 1, 3, 5, and 9) or a *c-Jun* expression plasmid (lanes 2, 4, 6, 7, 8, and 10) were assayed for the ability to bind CRE (lanes 1–8) and TRE (lanes 9 and 10) probes. Nuclear extracts were incubated with anti-c-JUN antibody (lanes 3 and 4) or anti-STAT5B antibody (lanes 5 and 6) and a 100-fold excess of unlabeled probe, CRE (lane 7) or STAT5B-binding probe (lane 8). Arrows indicate c-JUN specific complex and supershifted complex.

## Discussion

In the current study, we identified *Rcan1* as a gene induced by GH in HPX rat liver by using the SSH technique. We found that a bolus injection of GH rapidly and transiently induced *Rcan1-4* but not *Rcan1-1* transcription. GH also increased RCAN1-4 protein levels in HPX rat liver. Furthermore, GH caused the phosphorylation of c-JUN and transient induction of *c-Jun* mRNA. The results of luciferase and EMSA showed that the c-JUN promoted *Rcan1-4*

transcription through binding to the *Rcan1-4* promoter. Therefore, we suggest that c-JUN played a major role in the increase of *Rcan1-4* mRNA by GH.

*Rcan1-1* and *Rcan1-4* mRNAs have been analyzed by northern blot only in human tissues, and they were both 2.2-kb in length [37]. In addition to 2.2-kb *Rcan1* mRNA, a second transcript of about 2.0-kb *Rcan1* mRNA has been detected only in the liver [7], but it was not characterized because of a low amount of *Rcan1* mRNA. We found that GH increased *Rcan1-4* mRNA, but did not increase *Rcan1-1* mRNA. This may be caused by promoters of each mRNAs are different. GH increased the 3.0-kb and 2.0-kb *Rcan1* mRNAs in rat liver. The increased mRNAs are likely to be *Rcan1-4* mRNAs because GH did not increase *Rcan1-1* mRNA, which indicated that the levels of *Rcan1-4* mRNA were more abundant than those of *Rcan1-1* mRNA when GH existed. We also found that RCAN1-4 protein level in HPX rat liver was less than that of normal rat liver in both male and female rats. This suggests that GH regulates RCAN1-4 protein levels in normal rat liver. Anti-RCAN1 antibody detected 40 kDa protein, which corresponds to RCAN1-1 protein suggesting that small amount of RCAN1-1 protein existed in the HPX liver. RCAN1-1 protein levels were not changed by injection of GH as *Rcan1-1* mRNA were. Therefore RCAN1-4 protein seemed to be more abundant than RCAN1-1 protein when GH existed and almost equally scarce when GH did not exist. Because *Rcan1-4* mRNA is expressed in H4IIE hepatoma cells, it is expressed in hepatocyte, although it may be expressed in other liver cell types in the liver.

Previous studies have shown that expression of *Rcan1-4* is under the control of calcineurin. Calcineurin dephosphorylates transcription factor NFAT and dephosphorylated NFAT enter the nucleus and induce *Rcan1-4* mRNA through binding to clusters of NFAT binding sites upstream of exon 4 of *Rcan1* gene [19]. NFAT is known to interact with several transcription factors, such as AP-1 [18], GATA-2 [38], or MEF2 [39], to achieve gene transcription. Expression of *Rcan1-4* is regulated by NFAT in conjunction with GATA-2/3 in endothelial cells [35] and with C/EBPβ in myoblast cells [40]. *Rcan1-4* is also induced by endoplasmic reticulum (ER) stress. ATF6 induces *Rcan1-4* mRNA in ventricular myocytes [41], and c-JUN is responsible for induction of *Rcan1-4* mRNA in mouse fibroblast cells [41, 42] when ER stress is induced.

We showed that GH increased *Rcan1-4* mRNA levels independently of the calcineurin-NFAT signaling pathway in H4IIE GHR cells. We also showed that JNK inhibitor reduced the levels of GH induced *Rcan1-4* mRNA partially, which suggests that GH increases *Rcan1-4* mRNA levels through the JNK pathway. We showed that GH promoted phosphorylation of endogenous c-JUN and increased c-JUN protein level. Because JNK phosphorylates c-JUN at Ser63 and Ser73 to activate c-JUN [23], GH may activate JNK and promote phosphorylation of c-JUN. Ling *et al.* reported that GH stimulates JNK activity in NIH3T3 cells [43]. It has been reported that GH induces a small amount of nuclear *c-Jun* mRNA in HPX rat liver [44]. We found that a single injection of GH increased *c-Jun* mRNA levels vigorously. Activated c-JUN is known to autoregulate its promoter [27]. The profiles of *c-Jun* mRNA levels were consistent with the amount of Ser63-phosphorylated c-JUN protein, suggesting that the activated c-JUN increases *c-Jun* mRNA and increases total c-JUN protein by GH injection. The TRANS-FAC database identified one putative binding site designated as CRE and one designated as TRE upstream of exon 4 of *Rcan1* gene [36]. Luciferase assay showed that c-JUN regulated the expression of *Rcan1-4* mRNA, and EMSA showed that c-JUN bound to CRE but not TRE. These results suggest that GH increases the level of active c-JUN and induces *Rcan1-4* mRNA by binding of c-JUN to CRE (Fig 6). Lee *et al.* reported that pyrrolidine dithiocarbamate (PDTC), which elicits anti-inflammatory effects by inhibiting NF-κB signaling, induced *Rcan1-4* mRNA expression. They showed that PDTC also induced *c-Jun* and *c-Fos* mRNAs and that c-FOS bound to the region that includes CRE in a mouse macrophage cell line, suggesting that AP-1 induces *Rcan1-4* mRNA [45]. Because JNK inhibitor reduced the levels of

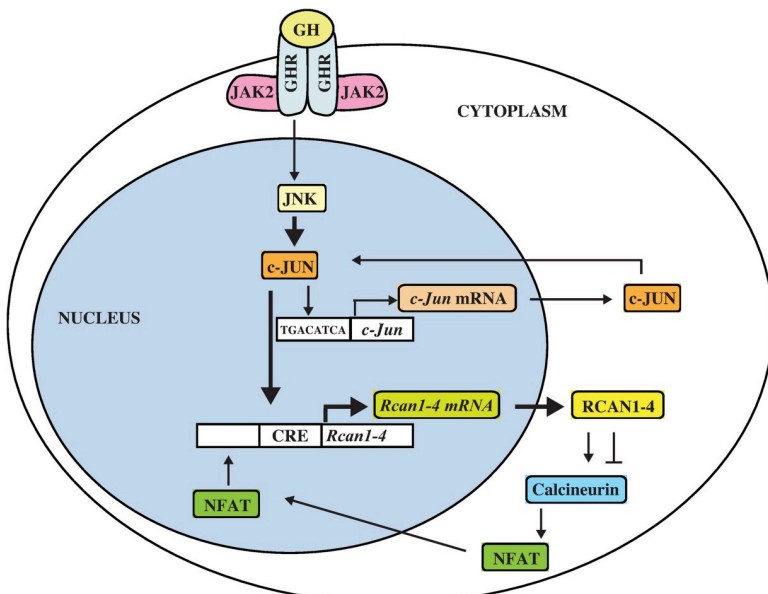

**Fig 6. Schematic illustration of GH signaling.** GH activates JNK, and activated JNK phosphorylates c-JUN. c-JUN binds to the CRE-like element (TGACATCA) in the promoter region of *c-Jun* and increases expression of *c-Jun*. Increased phosphorylated c-JUN binds to CRE upstream of exon 4 of *Rcan1* and increases *Rcan1-4* mRNA.

GH induced *Rcan1-4* mRNA partially, other unknown pathways are required for the induction of *Rcan1-4* mRNA by GH.

Experiments in different organisms and cell types have shown a dual function for RCAN1. Overexpression of RCAN1 inhibits calcineurin activity *in vitro* and *in vivo* [8, 9, 46, 47], whereas the results in *Rcan1* KO mice suggest that RCAN1 functions as a calcineurin facilitator [10, 11]. These results suggest that when RCAN1 level is low, it facilitates calcineurin activity, and when RCAN1 is high, it inhibits calcineurin activity. Phosphorylation also affects the function of RCAN1. In yeast, a low concentration of RCAN1 phosphorylated by GSK-3 stimulates the activity of calcineurin, and a high concentration of phospho- or dephospho-RCAN1 inhibits it [48]. TGF-beta-activated kinase 1 (TAK1) phosphorylates RCAN1 at Ser94 and Ser136, converting RCAN1 from an inhibitor to a facilitator of calcineurin-NFAT signaling in cardiomyocytes [49]. In our study, rat liver RCAN1-4 proteins migrated as a doublet in SDS gel electrophoresis, suggesting that two different phosphorylated states of RCAN1-4 proteins exist, but whether RCAN1-4 proteins increased by GH stimulate or inhibit the activity of calcineurin is under consideration.

RCAN1 functions in the immune response [50], angiogenesis [51], cardiac remodeling [9], and brain ischemia/reperfusion injury [52]. In the liver, RCAN1-4 acts as a potent tumor suppressor of HCC that attenuates tumor progression and angiogenesis *via* the inhibition of calcineurin-NFAT1 signaling. RCAN1-4 inhibits expression of vascular endothelial growth factor (VEGF), which is regulated by NFAT [53]. It has been reported that the serum level of VEGF is elevated in human GH deficient adults [54]. GH may regulate expression of VEGF by increasing RCAN1 and inhibiting calcineurin-NFAT signaling.

Recently, Seo *et al.* reported that RCAN1-4 protein levels were decreased in the livers of fasting mice, but increased in the livers of refed mice, mice fed a high fat diet (HFD), and *ob/ob* mice, suggesting that RCAN1-4 protein levels in the liver are related to nutritional status. They also showed that an extra copy of *Rcan1-4* elevated the expression levels of gluconeogenic genes and enhanced hepatic glucose production during fasting and impaired hepatic glucose

homeostasis, suggesting that RCAN1-4 participates in controlling hepatic gluconeogenesis [55]. GH may stimulate gluconeogenesis by increasing the RCAN1-4 protein levels.

In conclusion, since GH can activate multiple signal transduction pathways other than Stat5, we used subtraction hybridization to identify novel hepatic genes that may be regulated by these pathways, after a single injection of GH in HPX male rats. *Rcan1* was identified and the GHR signal transduction pathway further explored. We demonstrated that GH increases the expression of *Rcan1-4* and the levels of RCAN1-4 protein in rat liver. The active c-JUN, increased by GH, activates the promoter activity of *Rcan1-4* by binding to CRE. Further studies are needed to clarify the physiological significance of an increase of RCAN1-4 protein by GH.

## Supporting information

**S1 Table. GH-upregulated clones that were identified by virtual northern blot.**
(XLSX)

**S1 Data. Data set of figures.**
(XLSX)

**S1 Raw images.**
(TIF)

## Acknowledgments

We thank Ms. Mitsuko Kajita for sequencing the cDNAs.

## Author Contributions

**Investigation:** Tomoko Nakata, Yuka Toyoshima, Takashi Yagi, Harumi Katsumata, Reiko Tokita, Shiro Minami.

**Supervision:** Shiro Minami.

**Writing – original draft:** Tomoko Nakata.

**Writing – review & editing:** Shiro Minami.

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
