## [Decision Letter · Decision Letter 0]

13 Apr 2020

PONE-D-20-03557

Growth hormone increases regulator of calcineurin 1-4 (*Rcan1-4*) mRNA through c-JUN in rat liver

PLOS ONE

Dear Dr. Nakata,

Thank you for submitting your manuscript to PLOS ONE. After careful consideration, we feel that it has merit but does not fully meet PLOS ONE’s publication criteria as it currently stands. Therefore, we invite you to submit a revised version of the manuscript that addresses the points raised during the review process.

We would appreciate receiving your revised manuscript by May 28 2020 11:59PM. To enhance the reproducibility of your results, we recommend that if applicable you deposit your laboratory protocols in protocols.io, where a protocol can be assigned its own identifier (DOI) such that it can be cited independently in the future. For instructions see: http://journals.plos.org/plosone/s/submission-guidelines#loc-laboratory-protocols

We look forward to receiving your revised manuscript.

Kind regards,

Arun Rishi, Ph.D.

Academic Editor

PLOS ONE

Journal Requirements:

Reviewers' comments:

Reviewer's Responses to Questions

**Comments to the Author**

1. Is the manuscript technically sound, and do the data support the conclusions?

Reviewer #1: Yes

2. Has the statistical analysis been performed appropriately and rigorously? 

Reviewer #1: Yes

3. Have the authors made all data underlying the findings in their manuscript fully available?

Reviewer #1: No

4. Is the manuscript presented in an intelligible fashion and written in standard English?

Reviewer #1: Yes

5. Review Comments to the Author

Reviewer #1: This group used subtraction hybridization to identify hepatic genes that are regulated by a single GH injection in hypophysectomized male rats. This study focuses on the GH-mediated regulation Rcan1 showing the Rcan1-4, but not the Rcan1-1, is upregulated by GH. Evidence is provided this is likely through the induction of Jun-c.

The methods are solid, and for the most part the results are well described. The novelty of this work is based on the fact that we know very little regarding GH-mediated regulation of hepatic genes outside the vast literature focused on GHR-Jak2-Stat5b pathway.

Comments and Concerns

1) It is assumed that the subtraction hybridization method revealed multiple transcripts that were differentially regulated. This information should be provided to the general scientific community. A rationale should be included as to why the regulation of Rcan was pursued in detail. It appears that this same group used the subtraction hybridization data to also investigate GH-mediated XBP-1 that has recently published in Endocr Journal 2020.

2) Ref 1 refers to the somatomedin (IGF1) hypothesis and would be best used after information supplied in line 41. Consider other reviews for the multiple actions of GH.

3) On line 56, when NFAT first appears, please define acronym.

4) At the end of the discussion, the last sentence is not linked to preceding information about the multiple GHR mediated intracellular signals. Perhaps include more information. "Since GH can activate multiple signal transduction pathways other than Stat5, we used subtraction hybridization to identify novel hepatic genes that may be regulated by these pathways, after a single injection of GH in hypophysectomized male rats. Rcan1 was identified and the GHR signal transduction pathway further explored.

5) human GH is used in the in vitro studies. Since hGH can also bind to the prolaction receptor, is there evidence that this receptor is not expressed by the H4IIE cells. Also for this cell line indicated methods of maintenance, passage number ...

6) Since some of the methods may have been already reported in detail in the recent Endocr J 2020 on GH mediated xBP1 expression, they could be shorten and reference to the paper cited.

7) In the results section lines 243/244 indicates Rcan1 "Which has not been report as a GH responsive genes" How was this determined? Give references as to the manuscripts and data bases searched.

8) Fig 3 and 4 are qPCR and expressed relative to Veh treated. For hypox rat expression what is the relative level of Rcan1-1 vs -4 expression with or without GH? Do you think it is expressed in hepatcytes, or other liver cell types.

9) Not clear why figure ligands are put in main text, it was difficult to follow.

10) The beginning sentence of the abstract, might suggest the study will be looking more at the role of Rcan in liver function instead of how Rcan is regulated by GH, might consider starting abstract differently.

6. PLOS authors have the option to publish the peer review history of their article (what does this mean?). If published, this will include your full peer review and any attached files.

Reviewer #1: No

---

## [Author Response · Author response to Decision Letter 0]

26 May 2020

Response to Reviewers' Comments and Concerns

1) It is assumed that the subtraction hybridization method revealed multiple transcripts that were differentially regulated. This information should be provided to the general scientific community. A rationale should be included as to why the regulation of Rcan was pursued in detail. It appears that this same group used the subtraction hybridization data to also investigate GH-mediated XBP-1 that has recently published in Endocr Journal 2020. 

Reply

After screening clones by differential hybridization, clones were further screened by virtual northern blot and positive clones were sequenced. And then they were analyzed by northern blot. Therefore we added method of virtual northern blot in Method (lines 118-130). 

“Clones that hybridized specifically to the forward probe were screened by virtual northern blot analysis. Virtual northern blots were blots of full-length cDNA, made by PCR amplification of the primary cDNA [29]. cDNAs were amplified with Advantage 2 Polymerase Mix. Cycle conditions were 95 ˚C for 1 min, then 17 cycles of 95 ˚C for 5 sec, 65 ˚C for 5 sec, and 68 ˚C for 6 min in a Gene Amp PCR System 9700 (Applied Biosystems, Foster City, CA). The PCR product was separated on a 1.2% agarose gel, denatured and subsequently blotted onto a nylon membrane in a conventional southern transfer and UV-crosslinked. Each clone was labeled with PCR DIG Probe Synthesis Kit (Roche). Prehybridization, hybridization, washing and detection were performed according to the manufacturers' protocols. Clones that showed differential expression were sequenced by the BigDye Terminator Cycle Sequencing method using a 377 ABI Prism automated DNA sequencer (PerkinElmer) according to the manufacturers' protocols.”

We added data in S1Table and Fig 1A, and described a rational as follows (lines 216-229). 

“From the forward subtracted library (subtracting vehicle-treated cDNAs from GH-treated cDNAs), 702 colonies were isolated. 48 clones were hybridized with only forward subtracted probe but not reverse subtractive probe (subtracting GH-treated cDNAs from vehicle-treated cDNAs). To identify the clones that were really upregulated by GH, they were analyzed by virtual northern blot and 8 clones that were upregulated by GH were sequenced (S1 Table). mRNA levels of these genes were further determined by northern blotting to confirm that GH regulated the expression of these genes. We found that the levels of Rcan1, Xbp1, Nampt, and Leap2 mRNAs were much higher at 1 h after GH injection than those after vehicle injection (Fig 1A). These genes have not been described as GH responsive genes [31], although Wauthier and Waxman indicated in microarray data that Rcan1 and Nampt mRNAs were increased by GH [32]. We have studied Rcan1 and Xbp1 in detail because GH increased the levels of RCAN1 and XBP1 proteins. We reported about Xbp1 previously [28].” 

2) Ref 1 refers to the somatomedin (IGF1) hypothesis and would be best used after information supplied in line 41. Consider other reviews for the multiple actions of GH.

Reply

Ref 1 was moved to line 41 and replaced by the following review papers in line 38. 

1. Vijayakumar A, Novosyadlyy R, Wu Y, Yakar S, LeRoith D. Biological effects of growth hormone on carbohydrate and lipid metabolism. Growth Horm IGF Res. 2010;20(1):1-7. 

2. List EO, Sackmann-Sala L, Berryman DE, Funk K, Kelder B, Gosney ES, et al. Endocrine parameters and phenotypes of the growth hormone receptor gene disrupted (GHR-/-) mouse. Endocr Rev. 2011;32(3):356-86. 

3) On line 56, when NFAT first appears, please define acronym.

Reply

NFAT first appears on line 53 and we defined acronym (lines 52-53).

“Calcineurin dephosphorylates the nuclear factor of activated T cells (NFAT)” 

4) At the end of the discussion, the last sentence is not linked to preceding information about the multiple GHR mediated intracellular signals. Perhaps include more information. "Since GH can activate multiple signal transduction pathways other than Stat5, we used subtraction hybridization to identify novel hepatic genes that may be regulated by these pathways, after a single injection of GH in hypophysectomized male rats. Rcan1 was identified and the GHR signal transduction pathway further explored.

Reply

Thank you for the suggestion. According to your suggestion, we added the following sentences (lines 498-501).

“In conclusion, since GH can activate multiple signal transduction pathways other than Stat5, we used subtraction hybridization to identify novel hepatic genes that may be regulated by these pathways, after a single injection of GH in HPX male rats. Rcan1 was identified and the GHR signal transduction pathway further explored.”

5) human GH is used in the in vitro studies. Since hGH can also bind to the prolaction receptor, is there evidence that this receptor is not expressed by the H4IIE cells. Also for this cell line indicated methods of maintenance, passage number ...

Reply

Yes, human GH has lactogenic effects through binding to PRL receptor. Exclude the possibility, we studied the effect of PRL on the expression of Rcan1-4 in vivo, and the result was PRL did not increase Rcan1-4 mRNA significantly. We added the data in Fig 2D, and the following sentences (lines 259-261).

“Injection of the same amount of PRL did not significantly increase Rcan1-4 mRNA, which indicated that GH but not PRL increased Rcan1-4 mRNA (Fig 2D).”

We also added the following sentences in Methods (lines 149-152).

“H4IIE cells expressing the Ghr mRNA (H4IIE GHR) were constructed previously [28], and stored in alpha modified Eagle’s medium containing 10% fetal calf serum and 10% DMSO in liquid nitrogen freezer. H4IIE GHR cells at 16-20 passages were used in experiments.”

6) Since some of the methods may have been already reported in detail in the recent Endocr J 2020 on GH mediated xBP1 expression, they could be shorten and reference to the paper cited.

Reply

We shorten the methods.

7) In the results section lines 243/244 indicates Rcan1 "Which has not been report as a GH responsive genes" How was this determined? Give references as to the manuscripts and data bases searched.

Reply

Rcan1, Xbp1, Nampt, and Leap2 mRNAs have not been described as GH responsive genes in the paper of Flores-Morales et al (Flores-Morales A, Ståhlberg N, Tollet-Egnell P, Lundeberg J, Malek RL, Quackenbush J, et al. Microarray analysis of the in vivo effects of hypophysectomy and growth hormone treatment on gene expression in the rat. Endocrinology. 2001;142(7):3163-76.). Wauthier and Waxman provided the microarray data in the supplementary data on the web site of the paper (Wauthier V, Waxman DJ. Sex-specific early growth hormone response genes in rat liver. Mol Endocrinol. 2008;22(8):1962-74.), which indicated that Rcan1 and Nampt were GH responsive genes. Therefore we added their papers as the references and the following sentence (lines 223-229).

“We found that the levels of Rcan1, Xbp1, Nampt, and Leap2 mRNAs were much higher at 1 h after GH injection than those after vehicle injection (Fig 1A). These genes have not been described as GH responsive genes [31], although Wauthier and Waxman indicated in microarray data that Rcan1 and Nampt mRNAs were increased by GH [32]. We have studied Rcan1 and Xbp1 in detail because GH increased the levels of RCAN1 and XBP1 proteins. We reported about Xbp1 previously [28].”

8) Fig 3 and 4 are qPCR and expressed relative to Veh treated. For hypox rat expression what is the relative level of Rcan1-1 vs -4 expression with or without GH? Do you think it is expressed in hepatcytes, or other liver cell types.

Reply

We added these sentences in Discussion (lines 412-424).

“GH increased the 3.0-kb and 2.0-kb Rcan1 mRNAs in rat liver. The increased mRNAs were both likely to be Rcan1-4 mRNAs because GH did not increase Rcan1-1 mRNA, which indicated that the levels of Rcan1-4 mRNA were more abundant than those of Rcan1-1 mRNA when GH existed. We also found that RCAN1-4 protein level in HPX rat liver was less than that of normal rat liver in both male and female rats. This suggests that GH regulates RCAN1-4 protein levels in normal rat liver. Anti-RCAN1 antibody detected 40 kDa protein, which corresponds to RCAN1-1 protein suggesting that small amount of RCAN1-1 protein existed in the HPX liver. RCAN1-1 protein levels were not changed by injection of GH as Rcan1-1 mRNA were. Therefore RCAN1-4 protein seemed to be more abundant than RCAN1-1 protein when GH existed and almost equally scarce when GH did not exist. Because Rcan1-4 mRNA is expressed in H4IIE hepatoma cells, it is expressed in hepatocyte, although it may be expressed in other liver cell types in the liver.” 

9) Not clear why figure ligands are put in main text, it was difficult to follow.

Reply

We put the figure ligands in main text because of PLOS ONE's style requirements.

“Each figure caption should appear directly after the paragraph in which they are first cited.”

http://www.journals.plos.org/plosone/s/file?id=wjVg/PLOSOne_formatting_sample_main_body.pdf

10) The beginning sentence of the abstract, might suggest the study will be looking more at the role of Rcan in liver function instead of how Rcan is regulated by GH, might consider starting abstract differently.

Reply

We changed the beginning of the abstract as follows (lines17-20).

“Growth hormone (GH) activates multiple signal transduction pathways. To investigate these pathways, we identified novel genes whose transcription was induced by GH in the liver of hypophysectomized (HPX) rats using the suppression subtractive hybridization technique.”

---

## [Decision Letter · Decision Letter 1]

12 Jun 2020

Growth hormone increases regulator of calcineurin 1-4 (*Rcan1-4*) mRNA through c-JUN in rat liver

PONE-D-20-03557R1

Dear Dr. Nakata,

We’re pleased to inform you that your manuscript has been judged scientifically suitable for publication and will be formally accepted for publication once it meets all outstanding technical requirements.

Kind regards,

Arun Rishi, Ph.D.

Academic Editor

PLOS ONE

Additional Editor Comments (optional):

Reviewers' comments:

Reviewer's Responses to Questions

**Comments to the Author**

1. If the authors have adequately addressed your comments raised in a previous round of review and you feel that this manuscript is now acceptable for publication, you may indicate that here to bypass the “Comments to the Author” section, enter your conflict of interest statement in the “Confidential to Editor” section, and submit your "Accept" recommendation.

Reviewer #1: All comments have been addressed

2. Is the manuscript technically sound, and do the data support the conclusions?

Reviewer #1: Yes

3. Has the statistical analysis been performed appropriately and rigorously? 

Reviewer #1: Yes

4. Have the authors made all data underlying the findings in their manuscript fully available?

Reviewer #1: Yes

5. Is the manuscript presented in an intelligible fashion and written in standard English?

Reviewer #1: Yes

6. Review Comments to the Author

Reviewer #1: The authors have addressed all prior concerns and there are no additional concerns. .................

7. PLOS authors have the option to publish the peer review history of their article (what does this mean?). If published, this will include your full peer review and any attached files.

Reviewer #1: No

---

## [Editor Report · Acceptance letter]

18 Jun 2020

PONE-D-20-03557R1 

Growth hormone increases regulator of calcineurin 1-4 (*Rcan1-4*) mRNA through c-JUN in rat liver 

Dear Dr. Nakata:

I'm pleased to inform you that your manuscript has been deemed suitable for publication in PLOS ONE. Congratulations! Your manuscript is now with our production department. 

Kind regards, 

on behalf of

Prof Arun Rishi 

Academic Editor

PLOS ONE